# The Effect of Cannabis Plant Extracts on Head and Neck Squamous Cell Carcinoma and the Quest for Cannabis-Based Personalized Therapy

**DOI:** 10.3390/cancers15020497

**Published:** 2023-01-13

**Authors:** Kifah Blal, Elazar Besser, Shiri Procaccia, Ouri Schwob, Yaniv Lerenthal, Jawad Abu Tair, David Meiri, Ofra Benny

**Affiliations:** 1Department of Oral and Maxillofacial Surgery, Hadassah Medical Center, Faculty of Dental Medicine, The Hebrew University of Jerusalem, Jerusalem 9112102, Israel; 2Department of Pharmaceutical Science, School of Pharmacy, Faculty of Medicine, The Hebrew University of Jerusalem, Jerusalem 9112002, Israel; 3Laboratory of Cancer Biology and Cannabinoid Research, Department of Biology, Technion-Israel Institute of Technology, Haifa 3200003, Israel; 4Cannasoul Analytics Ltd., Caesarea 3079822, Israel

**Keywords:** apoptosis, cannabis, cannabinoids, cancer, HNSCC carcinoma, oral cancer, CBC, CBD

## Abstract

**Simple Summary:**

The survival rate of head and neck cancer has only improved slightly over the last quarter century, raising the need for novel therapies to better treat this disease. This research examined the anti-tumor effects of 24 different types of cannabis extracts on head and neck cancer cells. Type III decarboxylated extracts with high levels of Cannabidiol (CBD) were the most effective in killing cancer cells. From these extracts, the specific active molecules were recognized. Combining CBD with Cannabichromene (CBC) in a 2:1 ratio made the effect even stronger. These findings can help doctors match cannabis extracts to treat head and neck cancer. CBD extracts enriched with the non-psychoactive CBC can offer patients more effective treatment. Further research is needed to develop new topical treatments from such extracts.

**Abstract:**

*Cannabis sativa* plants have a wide diversity in their metabolite composition among their different chemovars, facilitating diverse anti-tumoral effects on cancer cells. This research examined the anti-tumoral effects of 24 cannabis extracts representative of three primary types of chemovars on head and neck squamous cell carcinoma (HNSCC). The chemical composition of the extracts was determined using High-Performance Liquid Chromatography (HPLC) and Mass Spectrometry (MS). The most potent anti-tumoral extracts were type III decarboxylated extracts, with high levels of Cannabidiol (CBD). We identified extract 296 (CAN296) as the most potent in inducing HNSCC cell death via proapoptotic and anti-proliferative effects. Using chemical fractionation of CAN296, we identified the CBD fraction as the primary inducer of the anti-tumoral activity. We succeeded in defining the combination of CBD with cannabichromene (CBC) or tetrahydrocannabinol (THC) present in minute concentrations in the extract, yielding a synergic impact that mimics the extract’s full effect. The cytotoxic effect could be maximized by combining CBD with either CBC or THC in a ratio of 2:1. This research suggests using decarboxylated CBD-type extracts enriched with CBC for future preclinical trials aimed at HNSCC treatment.

## 1. Introduction

Head and neck squamous cell carcinoma (HNSCC) [1] is the most common malignancy in the head and neck region. It develops from the mucosal epithelium in the oral cavity pharynx and larynx. HNSCC is the sixth most common cancer worldwide, accounting for approximately 900,000 cases and over 400,000 deaths annually as of 2020 [2,3,4,5]. Risk factors [6] include tobacco and alcohol consumption, exposure to environmental pollutants, and viral infection (HPV, EBV). Treatment modalities of confined HNSCC are resection, radiation, and systemic therapy. Despite advances in prevention, screening, HPV vaccination, immunotherapy, and combining treatment modalities, the 5-year survival rate of HNSCC only improved slightly from 55% to 66% between 1992 and 2006 when analyzed across all age groups and anatomic sites within the SEER registry [7]. Since the available treatments for HNSCC have been exhausted in many cases, there is a need to seek novel therapies for this malignant disease. Cannabinoids from the cannabis plant are promising therapeutic agents in cancer treatment, as preclinical studies demonstrated they could exert anti-tumor effects in different types of cancer cells [8,9]. 

Cannabis [10] is a genus of flowering plants in the Cannabaceae family [11], with *Cannabis sativa* as the main subspecies used in medicine due to its rich spectrum of bioactive phytochemicals, including phytocannabinoids, terpenes, and phenolic compounds. While the cannabis plant produces more than 500 different compounds, most of the focus has been on the two most abundant phytocannabinoids, the psychoactive molecule (-)-Δ^9^-trans-tetrahydrocannabinol (Δ^9^-THC) [12,13] and cannabidiol (CBD) [13]. However, studies in the last decade have demonstrated that other secondary metabolites present in cannabis extracts contribute to the overall therapeutic benefits of the plant [14] and are hypothesized to possess synergistic pharmacological properties with Δ^9^-THC or CBD [15], a phenomenon referred to as the “entourage effect” [16].

Phytocannabinoids’ effect on the human body is attributed mainly to their interaction with the endocannabinoid system (ECS) [17,18], an intricate signaling system distributed throughout the body, facilitating and regulating cellular level functions. The ECS mediates many physiological processes, including metabolism, cardiovascular regulation, reproduction, immune response, and pain sensation [19,20,21]. The ECS is composed of endogenous ligands, synthesizing and degrading enzymes, and features many cannabinoids receptors, which include the G-protein coupled receptors (GPCRs) cannabinoid receptor 1 (CB_1_) and cannabinoid receptor 2 (CB_2_) [22], orphan GPCRs including GPR12, GPR18, GPR35, GPR55, GPR119 [23], ligand-gated ion channels (i.e., transient receptor potential TRPV1, TRPV2, TRPA1, TRPM8) [24], and nuclear receptors (i.e., peroxisome proliferator-activated receptor-gamma, PPARγ) [25,26,27]. Cannabinoids were suggested as possibly beneficial for pathologies of the ears, nose, and throat via their modulation of the endocannabinoid system [28]. The role of the ECS in cancer generation and progression has not been entirely determined, with conflicting reports published [29,30,31,32,33]. However, with a multitude of published reports pointing to the role of the ECS in cell division and cancer development [9,30,32,34,35,36,37,38,39], there are many hopes for novel cannabis-based anti-cancer therapy [8]. In this research, we tested the effect of various cannabis extracts on HNSCC cells, characterized the chemical profile of those extracts with anti-tumoral activity, determined their anti-tumoral mode of action, tested their impact on non-cancerous cells, and examined endocannabinoid receptors involvement. 

## 2. Materials and Methods

### 2.1. Phytocannabinoid Extraction and Sample Preparation

Air-dried medical Cannabis cultivars were obtained from several Israeli medical Cannabis distributors. Cannabis extracts were prepared by The Laboratory of Cancer Biology and Cannabinoid Research, Department of Biology, Technion-Israel Institute of Technology, Haifa, Israel. Several samples were heat-decarboxylated in an oven at 130 °C for 1 h. Extracts were reconstituted in DMSO to achieve a 50 mg/mL concentration. For phytocannabinoid profiling, a fraction of the sample was diluted to achieve final concentrations of 1–10 μg/mL of Cannabis extract to DMSO. Pure synthetic Δ^9^-THC, CBD, CBN, and CBG were a kind gift from Pr. Rafael Mechoulam, Institute for Drug Research, Medical Faculty, Hebrew University, Jerusalem, Israel.

### 2.2. Phytocannabinoid Identification and Quantification

Phytocannabinoid analyses were performed using a Thermo Scientific ultra-high-performance liquid chromatography (UHPLC) system coupled with a Q ExactiveTM Focus Hybrid Quadrupole-Orbitrap MS (Thermo Scientific, Bremen, Germany). Identification and absolute quantification of phytocannabinoids were performed using analytical phytocannabinoid standards and Cannabis samples in a preset concentration by external calibrations as previously described [40].

### 2.3. Fractionation of the Whole Extract 

Fractionation of CAN296 into four fractions was performed using 1260 Infinity II LC System semi-preparative HPLC/UV (Agilent Technologies, Santa Clara, CA, USA). Fractions were collected in 10 min intervals and then lyophilized to dryness and analyzed by UHPLC/UV and ESI-LC/MS.

### 2.4. Reagents

Am630 (#1120/10), AMG9810 (#2316/10), and SET2 (6990/10) were purchased from Tocris Bioscience (Bristol, UK). BIM-46187 (#5332990001) and Rimonabant (#SML0800) were purchased from Sigma-Aldrich (Rehovot, Israel).

### 2.5. Cell Cultures

Four well-characterized human adherent epithelial cancer cell lines were purchased from the American Type Culture Collection (ATCC, Manassas, VA, USA): Scc4 (ATCC^®^ CRL1624^TM^), Scc9 (ATCC^®^ CRL1629^TM^), Scc25 (ATCC^®^ CRL1628^TM^), and Cal27 (ATCC^®^ CRL2095^TM^). A human adherent fibroblast cell line was purchased from the American Type Culture Collection (ATCC, Manassas, VA, USA): Hs895sk (ATCC^®^ CRL-7636^TM^). Cells were grown in DMEM/F-12 (Biological Industries, Kibbutz Beit-Haemek, Israel, 01-170-1A), supplemented with 2.5 mM L-glutamine (Biological Industries, 03-020-1A), 100 units/mL of penicillin G, 100 μg/mL of streptomycin (Biological Industries, 03-031-1B), 0.5 mM sodium pyruvate (Biological Industries, 03-042-1B) 400 ng/mL hydrocortisone (Sigma-Aldrich, 50-23-7) and 10% fetal bovine serum (Biological Industries, 04-007-1A). Cells were seeded and incubated to 70–80% confluence before experiments commenced. All cells were maintained in a humidified atmosphere at 37 °C with 5% CO_2_. 

### 2.6. Cell Viability Assays

Cells were seeded in 96-well plates at 7000 cells/well. Following overnight incubation, the growth media were discarded, and 100 µL/well fresh media containing 0.5% FBS were added to the cells. Different Cannabis extracts were added in triplicates at concentrations ranging from 1–10 μg/mL for 24 or 72 h, as indicated. DMSO was used as a control and applied in the same amount as the diluted extracts. Cells were then subjected to either MTT or WST-1 assays. For the MTT viability assay, 10 µL of MTT reagent (Sigma-Aldrich, M5655) was added to each well, and the cells were incubated at 37 °C for 3 h. Then, the medium was discarded, and 50 µL DMSO was added to each well. Plates were placed on an orbital shaker for 15 min to mix the contents, and plates were measured for absorbance using a microplate reader at OD = 590 nm. For the WST-1 assay, 10 µL of WST-1 (Abcam, Cambridge, UK, ab65473) was added to each well, and the cells were incubated at 37 °C for 1 h. Plates were placed on an orbital shaker for 15 min to mix the contents, and absorbance was read at OD = 440 nm. 

Cytotoxicity percentage was calculated with the following equation: (1)$$\%\;\mathrm{Dead}\;\mathrm{cells}=\frac{100\times \left(\mathrm{Control}-\mathrm{Sample}\right)}{\mathrm{Control}}$$

### 2.7. Cell Apoptosis Assay (Annexin V-APC/PI Assay)

Cells were seeded and treated with cannabis extracts as described for cell viability assays. Apoptosis was assessed by annexin V-APC (BioLegend, San Diego, CA, USA, 1640909/100) and PI staining in Annexin binding buffer (BioLegend, 421301/2) according to the manufacturer’s instructions. Ten thousand cells were acquired and analyzed using BD LSRFortessa^TM^ flow cytometer (BD, San Jose, CA, USA) and FlowJo software using FITC Annexin V Apoptosis Detection Kit with PI (BioLegend, San Diego, CA, USA). Results were calculated as the percentage of positive Annexin V-APC cells out of the total cells counted. 

### 2.8. Cell lysis and Western Blot Analyses

Following treatment with different concentrations of CAN296, Scc25, and Cal27, cells were solubilized in radioimmunoprecipitation assay buffer (Sigma-Aldrich, R0278), and protein concentration in lysates was determined using Bradford reagent (Sigma-Aldrich, B6916). Equal amounts of protein were resolved by NovexTM 4–20% Tris-Glycine Mini Gels (Thermo Fisher Scientific, Waltham, MA, USA, XP04200BOX) and electrophoretically transferred to a nitrocellulose membrane (Bio-Rad, Hercules, CA, USA, 1704159S). Membranes were blocked with Tris buffer saline (TBS) 0.1% Tween 20 buffer containing 5% BSA (Sigma-Aldrich, A7906) for one hr. The blots were then incubated overnight at 4 °C with an anti-cleaved caspase-3 antibody (Cell Signaling Technology, Danvers, MA, USA, 9664S), anti-cleaved PARP (Cell Signaling Technology, 5625) and GAPDH (Cell Signaling Technology, 5174). This was followed by incubation with horseradish peroxidase (HRP)-labeled with matching secondary antibodies. Immunoreactive bands were detected by Luminata^TM^ HRP substrate (Millipore, Burlington, MA, USA, WBLUR0500) and visualized using a MicroChemi imager (DNR Bioimaging Systems, Jerusalem, Israel). 

### 2.9. RNA Extraction

RNA was prepped using the DIRECT-zol kit from Zymo Research (Irvine, CA, USA) according to the manufacturer’s protocol. Briefly, a 10 cm plate grown to full confluency was washed twice with PBS and lysed with 1 mL TRI reagent (MRC—Molecular research center Inc.). The tubes were added 200 µL of chloroform and vortexed for 15 s before centrifugation at 11,000 RFC for 10 min. Approximately 600 µL of the aqueous upper layer was transferred to a clean tube, and ethanol was added 1:1 (*v*:*v*). The mix was loaded onto Direct-zol columns, washed, and eluted according to the manufacturer’s instructions, including DNAse treatment. Extracted RNA was run on Gel to ensure and quantified by Qubit. 

### 2.10. Real-Time Quantitative PCR

cDNA was prepared from 1 μg total RNA using the qScript cDNA kit (#95048-025, Quanta bio, Beverly, MA, USA) according to the manufacturer’s protocol. Relative mRNA expression levels of human receptors CNR1 (CB_1_), CNR2 (CB_2_), GPR55, TRPV1, TRPV2, TRPM8, and TRPA1 were quantified by qPCR in technical triplicates on a BIORAD CFX384 using the HOT FIREPol^®^ EvaGreen^®^ qPCR Mix (Solise Biodyne, Tartu, Estonia). As controls, two RNA mixes were used: MCF7 derived (which expresses a relatively low level of endocannabinoid receptors) and a mix of RNA derived from 3 cell lines—A549, DAUDI, and U-2OS expressing a high level of (expressing a relatively high level of endocannabinoid receptors). Ct results were normalized by calculating the triplicate Ct—Geometric mean Ct of 7 house-keeping genes: BRAP, CSNK, cul3, EIF4, GUSB, SF3, and STK-1G, which were selected based on RNA-seq data downloaded from The Cancer Cell Line Encyclopedia (https://sites.broadinstitute.org/ccle/, accessed on 26 February 2020) Housekeeping gene selection was based on three criteria. 1. Ubiquitous expression 2. Medium expression 3. Low CV (lowest variations between cell lines). Results are presented by calculating 2^−(∆-Ct)^ to allow linear comparison of gene receptor expression. All primer pairs were designed by NCBI’s primer-blast tools and tested for specificity by analyzing the dissociation curve for a single amplicon and comparing control results to the Cancer Cell Line Encyclopedia. 

### 2.11. Statistical Analysis

Statistical analyses were conducted using GraphPad Prism version 9.3.1 (GraphPad Software, LLC.). Data were reported as the mean ± SEM of at least three independent experiments. Multiple groups were compared using one-way, or two-way ANOVA followed by the Bonferroni post-hoc multiple comparisons test. A value of *p* ≤ 0.05 was considered significant for all tests. 

### 2.12. SynergyFinder 

Drug combination dose-response matrix data were analyzed with the SynergyFinder web application [41].

## 3. Results

### 3.1. The Heterogeneous Composition of Cannabis Extracts 

Twenty-four Cannabis extracts, natural (acid form, not decarboxylated) and after decarboxylation, were used for composition analysis. Using electrospray ionization liquid chromatography-mass spectrometry (ESI-LC/MS) [40], we quantified the phytocannabinoids profile of each extract. The 17 most abundant phytocannabinoids quantified in the 24 extracts are presented in the heatmap (Figure 1). A minimum concentration of 0.1% *w/w* was the inclusion criteria. The 24 extracts differed significantly in their phytocannabinoid content. Extracts were classified into three groups according to their THC and CBD content [42] as follows: THC-type with high THC (~50% *w*/*w*) and low CBD concentration (<1% *w*/*w*), CBD-type with high CBD (~50% *w*/*w*) and low THC concentrations (<3% *w*/*w*), and THC:CBD group with equal (~30% *w*/*w*) THC and CBD concentrations. Further, each of the three groups was divided into two subgroups, decarboxylated and natural. 

### 3.2. Decarboxylated High-CBD Extracts Are the Most Effective in Inducing HNSCC Cell Death

To examine the effect of the 24 extracts on the survival of HNSCC, we used four well-established oral origins of SCC cell lines: Scc4, Scc9, Scc25, and Cal27. We treated cells with increasing concentrations ranging from 2–10 μg/mL of all 24 cannabis extracts for 24 h (Appendix A). Cell viability was evaluated using the MTT cell viability assay, and the percentage of dead cells was calculated. Of the six groups of clusters, extracts of the decarboxylated CBD-type significantly decreased the cell viability of all four cell lines. A representative comparison at a concentration of 6 µg/mL is presented in Figure 2A. 

Based on the initial screening, all extracts in the decarboxylated CBD-type group and one extract in the decarboxylated THC:CBD-type group were significantly effective against HNSCC cancer cells relative to the other extracts. The decarboxylated CBD-type CAN296 extract was chosen to further characterize this group of extracts. An additional study was conducted to reconfirm and examine the effect of increasing concentrations of new batches of CAN296 extract on the cell survival of the four cell lines tested. We measured the survival of the four cell lines after 24 h of treatment using the MTT cell viability assay. A dose-dependent response was observed (Figure 2B), and the IC50 ranged between 4–6 µg/mL depending on the type of cell line treated, with no significant differences between the four cell lines. As no significant differences were observed between the cell lines, Scc25 and Cal 27 were chosen for further investigation as representative cell lines for HNSCC cancer cells. 

### 3.3. Cannabis Extract CAN296 Shows Apoptotic Effects in HNSCC

To examine the mechanism of cancer cell death induced by the cannabis extract, we treated Scc25 and Cal27 for 12 h with CAN296 extract at concentrations of 2–8 µg/mL. Then, cells were stained for Annexin V-APC/PI and assessed for apoptosis with single-cell analysis by flow cytometry (FACS) (Figure 3A,B). Treatment resulted in increased cell death through apoptosis, both early and late, in a dose-dependent manner. To further verify the induction of apoptosis, we assessed cleaved Caspase-3 and PARP with western blot analyses, and findings confirmed a dose-dependent induction of cleaved Caspase-3 and PARP following treatment with CAN296 extract (Figure 3C and Appendix A). Taken together, these results suggest the cannabis extract induces programmed cell death of the HNSCC cells via apoptosis in a dose-dependent manner. We examined the prolonged effect of the Cannabis extracts on HNSCC by comparing Scc25 cells treated with CAN296 for either 24 or 72 h. Cells were assessed via WST-1 proliferation assay. The cannabis extract had a significant anti-proliferative effect on Scc25 at 4–6 µg/mL when applied for 72 h. At 8 µg/mL, the extract was already effective after 24 h (Figure 3D).

### 3.4. Cannabis Extract CAN296 Is More Selective Towards HNSCC Cells Compared to Non-Cancer Fibroblastic Cells 

We examined the selectivity of extract CAN296 to HNSCC cell lines by comparing the normal skin fibroblast cell line Hs895sk to Scc25 cells when treated with increasing concentrations (2, 4, 6, and 8 µg/mL) of CAN296 (Figure 4). While Scc25 cell death was significantly increased at 6 and 8 µg/mL after 24 h of treatment, Hs895sk cell viability was unchanged. At 72 h, Hs895sk cells were resistant to the apoptotic and anti-proliferative effects of CAN296 in concentrations up to 6 µg/mL. Treatment with 8 µg/mL for 72 h resulted in increased cell death of Hs895sk relative to 24 h of treatment, approximately 55%; however, Scc25 were significantly more sensitive as treatment resulted in significantly higher percentages of cell death, approximately 90%.

### 3.5. Endocannabinoid Receptors Expression in HNSCC Cells 

To better understand the mode of action of Cannabis extracts on HNSCC cells, we analyzed the mRNA levels of seven common endocannabinoid receptors in the four cell lines using real-time qPCR. Three of the cell lines expressed high levels of *CNR1* (CB_1_), and all except Scc4 demonstrated very low expression of *GPR55*. Receptors *CNR2*, *TRPA1*, and *TRPM8* were expressed in a mixed pattern across the cell lines. *TRPV1* was the primary receptor expressed in all the cell lines, and *TRPV2* was highly expressed in Scc4 and moderately expressed in all other cell lines tested (Figure 5A).

To identify endocannabinoid receptors’ involvement in the cytotoxic activity of the cannabis extract, we pretreated Scc25 cells with different inhibitors: the inverse agonists (IA) to CB1 (Rimonabant) and CB2 (AM-630), the two receptors of the classic ECS, and the general inhibitor of Gαq-proteins (BIM-46174); as well as antagonists (ant.) to TRPV1 (AMG 9810) and TRPV2 (SET2) that were highly expressed. None of the inhibitors significantly rescued the cells from the cytotoxic effect of extract CAN296. (Figure 5B).

### 3.6. Identifying the Active Molecules in CAN296 Responsible for the Apoptotic Effect

To identify which active molecules in the CAN296 extract were responsible for the apoptotic effect on the HNSCC cancer cells, preparative Ultra-high-performance liquid chromatography (UHPLC) was used to fractionate CAN296 extract into four distinctive fractions (Figure 6A). Each of the four fractions had a unique dominant cannabinoid content; Fraction 1 was CBDV dominant, Fraction 2 was CBD dominant, Fraction 3 was THC dominant, and Fraction 4 was CBC dominant (Figure 6B). 

The fractions were normalized to match their concentration in the whole extracts (*w*/*w*) by diluting each fraction to match its dominant cannabinoid concentration to its concentration in the whole extract, and Scc25 cells were treated with each fraction or different combinations. Cells were treated in increasing overall concentrations (concentrations 4, 6, 8 µg/mL) for 24 h and analyzed for cell viability (Figure 7). Fraction 2 (containing CBD) induced significant cell death at 8 µg/mL. When combined with either Fraction 3 (containing THC) or Fraction 4 (containing CBC), cell death was significant already at 6 µg/mL.

### 3.7. Combination of CBD with CBC or THC Results in a Synergistic Effect on HNSCC Cell Viability

To better understand the relationship between CBD, CBC, and THC treatment, i.e., whether there is an additive or synergistic correlation, we tested combinations of synthetic cannabinoids CBD, THC, and CBC. We designed a study of a dose-response matrix for CBD/CBC and CBD/THC and examined their effect on the viability of Scc25 cells. Results were analyzed for Bliss synergy [41] and were plotted. We found a synergy correlation between CBD/CBC (synergy score, 28.01) at a 2:1 concentration ratio (Figure 8A) and between CBD/THC (synergy score, 15.41) at a 4:3 concentration ratio. (Figure 8B).

We further examined the effect of increasing concentrations of 2–8 µg/mL (total cannabinoid concentration) of CBD and CBC combination at the synergistic ratio of 2:1 relative to CAN296 on Scc25 cells survival. Cell viability was evaluated using the MTT cell viability assay. We found that the synergistic combination of CBD and CBC had a significantly greater effect on cell viability than the whole extract with an IC50 of 2 µg/mL compared to 4 µg/mL in the whole extract (Figure 8C).

## 4. Discussion

In this research, we compared the cytotoxic effect of 24 representative cannabis extracts [42] on HNSCC cells. We found that decarboxylated CBD-type Cannabis extracts led to a significant increase in cell death, and this effect was consistent throughout all HNSCC cell lines tested (Scc4, Scc9, Scc25, Cal27). The decarboxylated CBD-type CAN296 extract was chosen for further characterization. We found the cytotoxic effect to be dose-dependent, acting via the induction of apoptosis. Moreover, the cytotoxic effect was specific to HNSCC cancer cells, as similar concentrations did not induce apoptosis in fibroblasts, falling in line with previous reports indicating the selectivity of the apoptotic effect by Cannabis extracts to cancer cells [8,43,44,45].

By nature, cannabis extracts display a wide variance in their metabolomic composition, with conflicting reports of the anti-tumoral effect [45,46,47]. Consequently, active plant ingredients (APIs) must be identified and standardized for adequate clinical anti-cancer treatment. For that purpose, the chosen crude extract CAN296 was fractionated, and different combinations were formed, normalized, and tested for cytotoxicity on Scc25 cells. While the CBD dominant fraction was able to induce a cytotoxic effect on the cancer cells by itself, potency was notably less than the whole extract, emphasizing that the cytotoxic effect of decarboxylated CBD-type whole extract on HNSCC cells is higher than pure CBD. Moreover, the group of extracts with the highest CBD content (decarboxylated CBD-type) possessed the highest cytotoxic effect, extracts with moderate CBD content (decarboxylated THC:CBD-type) had a moderate cytotoxic effect, as seen with extract CAN298, and extracts with minimal CBD (all other groups) had a neglectable cytotoxic effect.

When the CBD dominant fraction was combined with any of the CBC or THC fractions, the cytotoxic effect matched that of the whole extract, pointing out an entourage effect between CBD-CBC and CBD-THC, a known feature of cannabis extracts [16]. Interestingly, in the decarboxylated CBD-type group, both CBC and THC were present in low amounts of 2.5–3.3%, with a ratio to CBD of 1:15–1:20, respectively. Higher amounts of both molecules were needed to achieve the full cytotoxic potential equivalent to that of the extract. Synergy model testing confirmed the combination between CBD-CBC or CBD-THC is synergistic, with an optimal ratio of 2:1, respectively.

While previous reports of THC-CBD synergy demonstrated reduced tumor growth [39,48,49], some studies suggested combining THC with CBD mainly modulated the adverse effects of THC [13,50,51]. This research points out CBD as the main API, and combining it with either CBC or THC at a ratio of 2:1 maximizes the cytotoxicity on HNSCC cells. Considering the adverse psychotomimetic effects of THC, there is a clear advantage for favoring the CBD-CBC combination over CBD-THC for novel treatments for HNSCC.

To the best of our knowledge, only one previous report [52] pointed to the interesting biological activity of CBC and its potential for medical use. With the mechanism yet to be discovered, a synergy between CBC was found to enhance the cytotoxic effect of THC in treating Urothelial Cell Carcinoma [52]. Our research found CBC to enhance the cytotoxic effect of CBD, establishing additional support for the phenomenon of the entourage effect in phytocannabinoids [53,54,55].

CBD has been previously reported to affect cancer cell growth through different pathways, including the induction of cell cycle arrest at the G_0_-G_1_ phase, increasing the levels of proapoptotic signaling proteins such as BSD, BAX, and ROS [56], increasing the levels of cleaved caspase-3 and -9, or by activating the mitochondrial apoptotic pathway [56,57]. The combination of CBD with CBC was more effective than CBD by itself. CBD can bind multiple targets, and CBC was shown to bind TRP [58] channels. Our finding aligns with publications [24,58] that demonstrated that some Cannabis extracts enriched with specific cannabinoids were more potent agonists of TRPA1, TRPV1, TRPV2, and TRPM8 receptors compared to these same pure phytocannabinoids individually. Yet, it is not clear whether the cytotoxic effect of CBD and CBC combination is the result of one cannabinoid enhancing the activity of the other or from hitting different signaling pathways at once.

We examined RNA expression in order to characterize our relevant cell lines in terms of gene expression of cannabinoid receptors [22], as well as receptors known to mediate cannabinoid signaling [23,24]. A high expression of *CNR1* (CB_1_), *TRPV1*, and *TRPV2* was noted. In contrast, the expression of *CNR2* (CB_2_) was very low, which falls in line with previous works suggesting that CB_2_ is expressed mainly by immune cells [22]. *GPR55*, *TRPA1*, and *TRPM8* receptors expression was not consistent in all cell lines and ranged from medium to low. While CBD is an antagonist or a negative allosteric modulator of CB_1_ [59,60], CBD and THC can activate TRPV2, TRPA1, and TRPM8 receptors [24,61], and both CBD and CBC have been recorded to activate TRPA1 [62]. Nevertheless, in this study, all attempts to block the cytotoxic activity of CAN296 extract by inhibiting CB1, CB2, Gq, GPR55, TRPV1, and TRPV2 failed to reduce the cytotoxicity of CAN296 treatment, suggesting that the mode of action of the extract on HNSCC does not involve these specific receptors.

## 5. Conclusions

This research suggests using whole cannabis extracts, which are decarboxylated CBD-rich, to induce cancer cell death. In recent years many cancer patients have been treated with cannabis extracts as a palliative treatment. Based on this research, these chemovars can provide additional anti-cancer properties in addition to the palliative effects that cancer patients can benefit from. Furthermore, we recommend enriching extracts with CBC to reach a CBD to CBC ratio of (2:1) to maximize the cytotoxic effect on HNSCC.

Further research is mandatory on ex-vivo cancer models of HNSCC and in-vivo studies to compare the local and systemic effects of decarboxylated CBD-rich cannabis extract to a single cannabinoid combination of CBD CBC. Developing a topical mucoadhesive formula for the future application of cannabis extracts and single cannabinoids to the oral mucosal cancer lesion is still in its early stages.

## Figures and Tables

**Figure 1 cancers-15-00497-f001:**
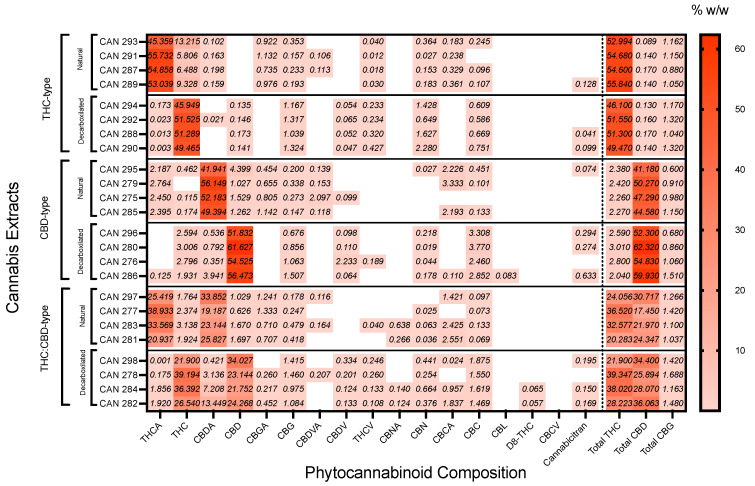
Heatmap clustering of the cannabinoid profile of 24 cannabis extracts. Matrix of the ESI-LC/MS composition. Phytocannabinoid analysis scores were scaled by column to range from 0 to 100. A low score (pale red) indicates that the %*w*/*w* ratio of the cannabinoid in the extract is meager compared to its percentage in the other extracts. A high score (darker red) indicates a high ratio in the extract compared to its percentage in the other extracts. The 24 extracts are segregated into six major clusters: THC-type (dominant THC over CBD content) decarboxylates/acid form, CBD-type (dominant CBD over THC content) decarboxylates/acid form, THC:CBD-type (equal THC and CBD content) decarboxylates/acid form.

**Figure 2 cancers-15-00497-f002:**
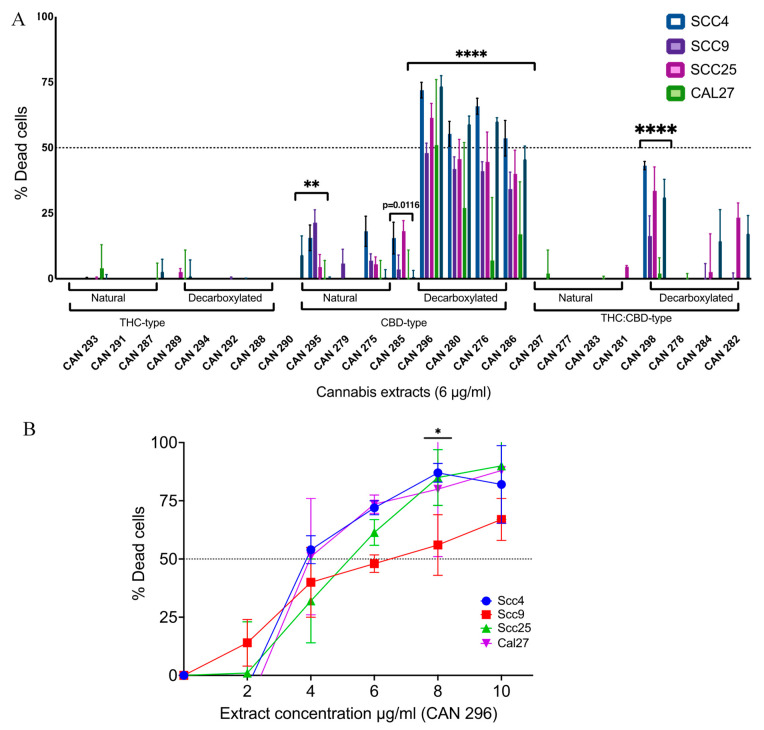
Differential effect of different Cannabis extracts on the viability of HNSCC cells. (**A**) Cell viability of HNSCC cell lines Scc4, Scc9, Scc25, and Cal27 following treatment with 24 cannabis extracts at 6 µg/mL concentration for 24 h. Cell viability was evaluated by MTT, and data are reported as mean ± SE (n = 3) of % dead cells compared to DMSO control, $\%\;\mathrm{Dead}\;\mathrm{cells}=\frac{100\;\times \;\left(\mathrm{Control}\;-\;\mathrm{Sample}\right)}{\mathrm{Control}}$. Differences were statistically analyzed with two-way ANOVA followed by Sidak’s multiple comparisons test (** *p* < 0.001, **** *p* < 0.0001). (**B**) Dose-response curve of cell lines Scc4, Scc9, Scc25, and Cal2, following 24 h of treatment with CAN296 (decarboxylated CBD-type) extract. Data are reported as mean ± SE (n = 3) of % Dead cells compared to DMSO control. Statistically analyzed with two-way ANOVA followed by Tukey’s multiple comparisons (* *p* < 0.05).

**Figure 3 cancers-15-00497-f003:**
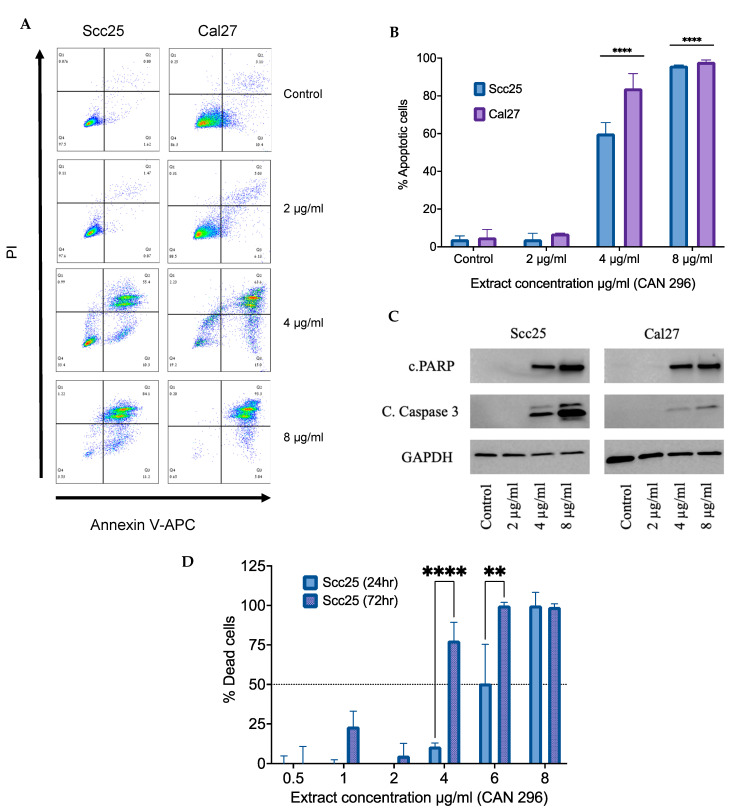
Apoptotic effect of Cannabis extracts on HNSCC. Extract CAN296 (2, 4, 8 µg/mL) was applied on Scc25 or Cal27 cells for 12 h, with DMSO as the control. (**A**) Apoptosis (early and late) was assessed by APC Annexin-V/PI staining with flow cytometry. (**B**) The percent of apoptotic cells was calculated as % of positive Annexin-V APC cells out of the total cells counted (events = 10,000) and presented as mean ± SE (n = 3). Statistically analyzed with two-way ANOVA followed by Tukey’s multiple comparisons test, and asterisks indicate significant differences compared to the control (**** *p* < 0.0001). (**C**) Cells were lysed and resolved on 15% SDS-PAGE, followed by western blotting with anti-Cleaved Caspase 3, c.PARP and GAPDH as the loading control. (**D**) Scc25 cells were treated with 0.5–8 µg/mL CAN296 extract for either 24 or 72 h, and cell proliferation was assessed according to WST-1 assay. The percent of vital cells relative to DMSO (control) is presented as mean ± SE (n = 3). Asterisks indicate statistical significance between 24 h and 72 h treatment (** *p* < 0.0005, **** *p* < 0.0001; two-way ANOVA with Sidak’s post-hoc multiple comparison test).

**Figure 4 cancers-15-00497-f004:**
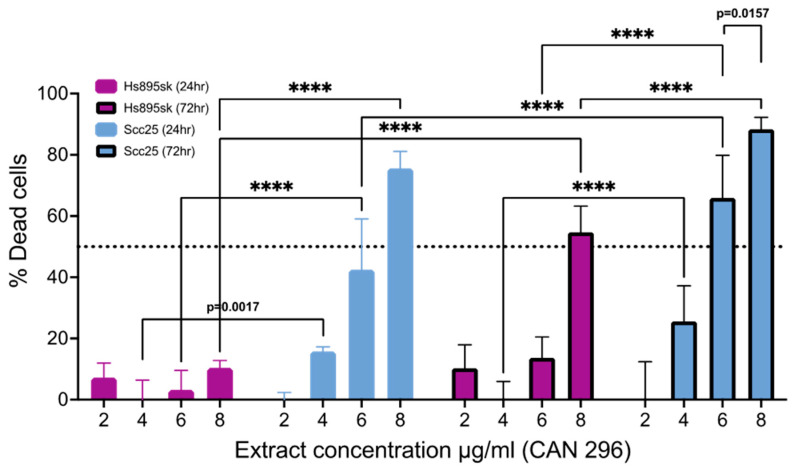
Cannabis extract CAN296 is selective to HNSCC. Scc25 and Hs895sk cell lines were treated with CAN296 extract at 2, 4, 6, or 8 µg/mL for 24 or 72 h. Cell viability was assessed via WST-1 cell toxicity assay. The percent of dead cells relative to the DMSO control is presented as mean ± SE (n = 3). Statistically analyzed by two-way ANOVA with Tukey’s multiple comparison test (**** *p* < 0.0001).

**Figure 5 cancers-15-00497-f005:**
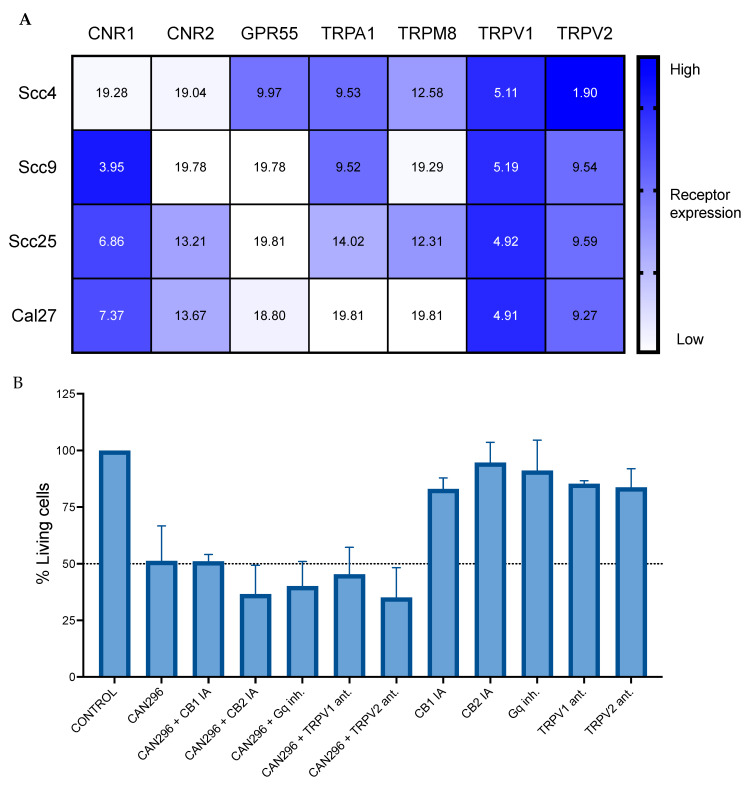
The cytotoxic effects of CAN296 are not mediated via common ECS receptors. (**A**) CNR1 (CB1), CNR2 (CB2), GPR55, TRPA1, TRPM8, TRPV1, and TRPV mRNA levels were evaluated by qPCR and normalized to the geomean of 7 house-keeping genes: BRAP, CSNK, cul3, EIF4, GUSB, SF3, and STK-1G. Expression levels are represented as ΔCT and color-coded (lower ΔCT values indicate higher receptor expression). Results are presented as a mean expression (n = 3). (**B**) Scc25 cells were left untreated or pretreated with endocannabinoid receptors antagonists for 1 h: 10 µM Rimonabant—CB1 (IA), 10 µM Am630—CB2(IA), 10 µM BIM46187—Gq (inh.), 20 µM AMG9810 TRPV1(ant.), or 20 µM SET2—TRPV2 (ant.). Then, cells were treated for 24 with 4 µg/mL CAN296 extract, and cell viability was evaluated by MTT relative to DMSO control. Data are reported as mean ± SE of % viable cells out of DMSO control untreated cells (n = 6).

**Figure 6 cancers-15-00497-f006:**
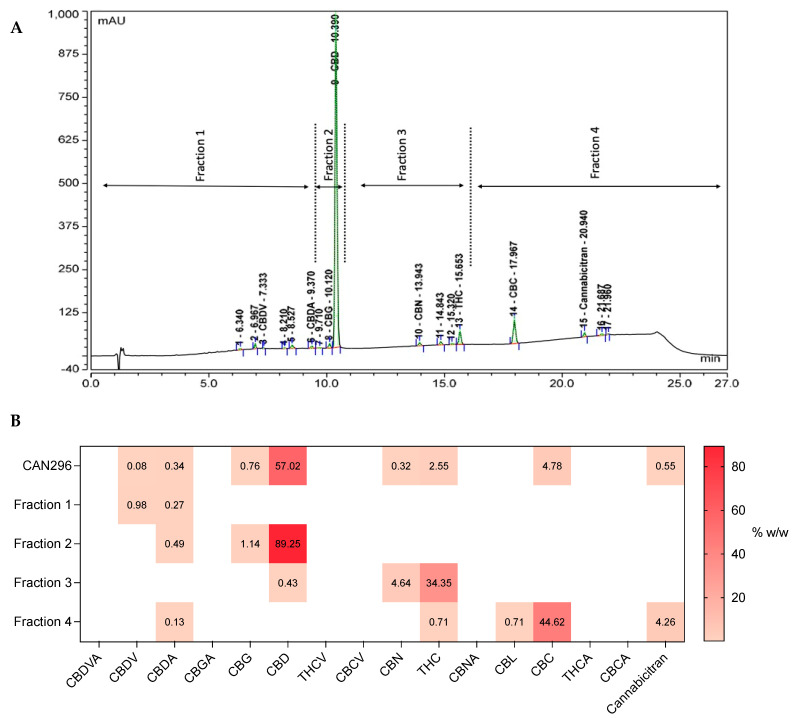
Chromatographic separation & heatmap clustering of the cannabinoid content of CAN296 and its fractions 1–4. (**A**) UHPLC chromatogram of cannabis extract CAN296; vertical lines indicate the retention time cutoff: fraction 1, 0–9.36 min; fraction 2, 9.36–10.393 min; fraction 3, 10.42–15.653 min; fraction 4, 15.657–21.963 min. (**B**) Heatmap of CAN296 and fraction 1–4 cannabinoids content. The values present the content percent *w/w* of each of the tested cannabinoids in the whole extract and each fraction according to ESI-LC/MS analysis. Higher scores (darker red) indicate a higher percentage of the cannabinoid.

**Figure 7 cancers-15-00497-f007:**
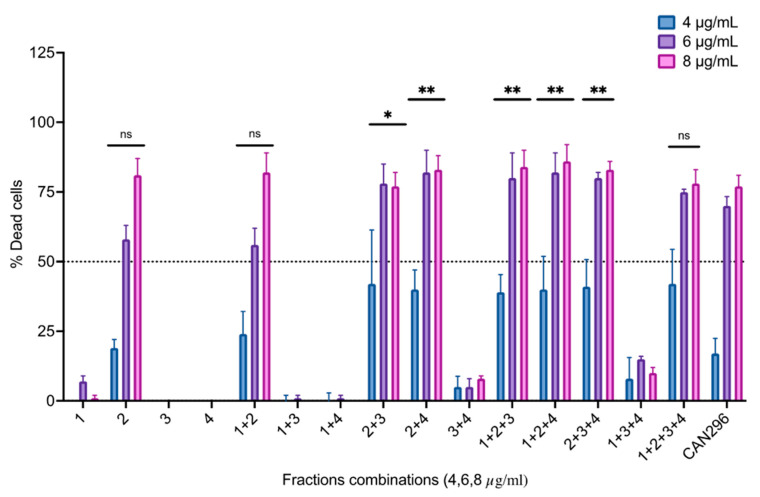
Differential effects of different fraction combinations on the survival of HNSCC cells. The oral Scc25 cell line was treated with 4, 6, and 8 µg/mL of each fraction or combinations of fractions for 24 h, and cell viability was assessed via MTT assay relative to the DMSO-treated control. Data are reported as mean ± SE (n = 3) of % dead cells out of the total cells. Asterisks represent statistically significant differences compared to CAN296 (* *p* < 0.05, ** *p* < 0.01, two-way ANOVA with Dunnett’s multiple comparisons test).

**Figure 8 cancers-15-00497-f008:**
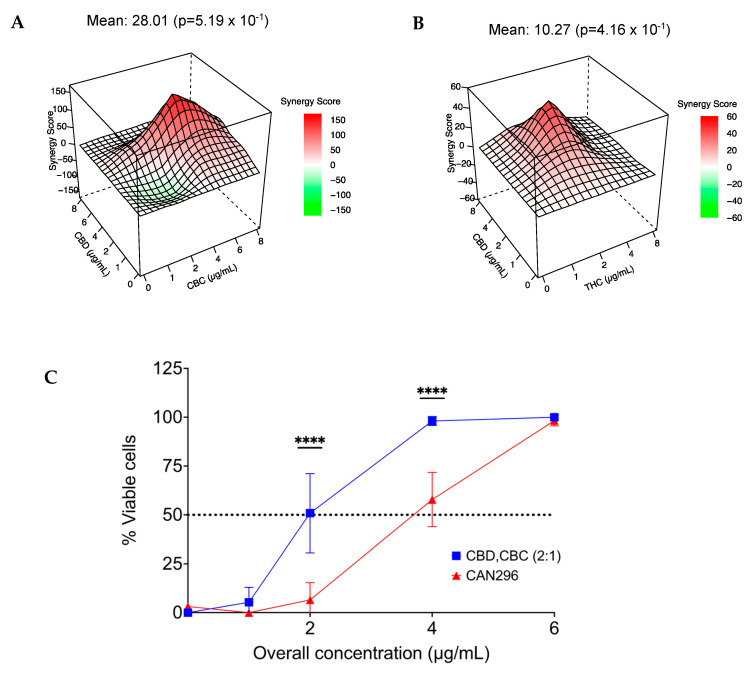
CBD and CBC combination (2:1 ratio) is more effective than CAN296. The Bliss model synergy distribution for (**A**) CBD-CBC, and (**B**) CBD-THC, was calculated with the SynergyFinder web application (v 2.0). Data are reported as a 3D Visualization synergy map in a bliss synergy mode, with multiplicative effect as if the two cannabinoids act independently. Cell death was assessed by MTT cell viability assay (n = 3). (**C**) A dose-response curve of cell line Scc25 following 24 h of treatment with CAN296 extract vs. treatment with CBD and CBC cannabinoid combination at the synergistic ratio of 2:1. Data are reported as mean ± SE (n = 6) of % viable cells compared to control. Asterisks indicate statistical significance in cell viability between treatments (**** *p* < 0.0001, two-way ANOVA with Sidak multiple comparison tests).

## Data Availability

The data underlying this article will be shared upon reasonable request to the corresponding author.

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
