# Peer review of "The Effect of Cannabis Plant Extracts on Head and Neck Squamous Cell Carcinoma and the Quest for Cannabis-Based Personalized Therapy"

_cancers, 2023, doi:10.3390/cancers15020497_

Round 1

Reviewer 1 Report

thank you for the opportunity to read and comment this paper

This is an interesting  “ in vitro study”  that could represent the basis of further clinical use of the cannabis plant ectracts.

The Authors stated on page 2 that “ Cannabinoids from cannabis plant are promising therapeuthic agents in cancer treatment “ this is true if we consider their effects on pain, anxiety, depression, appetite  or general well being. On the contrary at the best of my knowledge till now is not well documented any direct effect of cannabis on cancer it self. This also is the conclusion of a recent review by Caputo and  Rodriguez (2021)

The Authors focused their study on head and neck cancer cells as a consequence I suggest (on page 2) to consider the recent paper by Brandon Tapasak & coll on “endocannabinoid System and the otolaryngologist” (Clin N Am 2022).

I think the Authors should be more cautious in transferring their in vitro studies into clinical practice.

Also in the conclusion on page 15 the  Authors show an optimism that I would like to share. Unfortunately the clinical data do not support any conclusions on utility/ safety of  cannabis/cannabinoids as an adjuntive treatment in head neck cancer . Moreover , even the routes of administration should be better studied in the future being some of these not approved  by FDA.

With the utmost respect for the authors , In my opinion the last tip about the development of a topical adhesive for oral cancer treatment represents a good dream rather difficult to realize.

All the best

mg

Author Response

Reviewer 1

The Authors stated on page 2 that “ Cannabinoids from cannabis plant are promising therapeuthic agents in cancer treatment “ this is true if we consider their effects on pain, anxiety, depression, appetite  or general well being. On the contrary at the best of my knowledge till now is not well documented any direct effect of cannabis on cancer it self. This also is the conclusion of a recent review by Caputo and  Rodriguez (2021)

Au: In addition to the effects of cannabinoids mentioned by the reviewer, in preclinical studies cannabinoids have been shown to possess anti-tumor properties and their potential as anticancer agents dates back two decades. We reference almost ten papers from the field showing anticancer effects of cannabinoids on different cancer types, including a reference to a paper by our group that tested cell lines representative of different cancer types (Baram et al., 2019). As these references appear later on in the manuscript, we added a clarification to the sentence to clarify why cannabinoids are promising therapeutic agents with adequate references.

The Authors focused their study on head and neck cancer cells as a consequence I suggest (on page 2) to consider the recent paper by Brandon Tapasak & coll on “endocannabinoid System and the otolaryngologist” (Clin N Am 2022).

Au: This recent paper suggest cannabinoids are possibly beneficial for pathologies of the ears, nose, and throat via their modulation of the endocannabinoid system. We added a reference to this paper in the final paragraph of the introduction.

I think the Authors should be more cautious in transferring their in vitro studies into clinical practice.

Au: We agree with the reviewer it is important to be cautious when interpreting in vitro results to clinical settings, and we stress further research is needed in the conclusion.

Also in the conclusion on page 15 the  Authors show an optimism that I would like to share. Unfortunately the clinical data do not support any conclusions on utility/ safety of  cannabis/cannabinoids as an adjuntive treatment in head neck cancer . Moreover , even the routes of administration should be better studied in the future being some of these not approved  by FDA.

With the utmost respect for the authors , In my opinion the last tip about the development of a topical adhesive for oral cancer treatment represents a good dream rather difficult to realize.

Au: We apologize for an incomplete sentence in the end of the conclusion section that may have caused confusion. We agree with the reviewer there is still a long road ahead before cannabinoids can be used as anticancer treatments, and the correction we made should convey that.

Reviewer 2 Report

In this paper, the authors analyse the anti-tumour effects of a series of 24 different types of cannabis extracts on head and neck cancer cells.

The authors determine which is the best of the extracts examined and show that the richest in CBD are the most effective. However, the CBD molecule alone is no better than the most effective extracts. If there were other molecules present in the mixture, they would act synergistically. The bottom line is that more research is needed.

In my opinion, it is an excellent work that brings a new perspective on the potential of cannabis extracts in cancer therapy.

The sentence “Yet, CBD and CBC may act together by alternative pathways independent of receptor mediation” is not clear. A cannabinoid can be an agonist of a receptor, two at the same time not, they will compete and act depending on the affinities and concentrations according to the law of mass action. I imagine the authors mean that there are phytocannabinoids in the extract that act on different receptors. Do the authors have more information than what appears in reference 22?

The sentence “The suggested combination of CBD-CBC in this research aligns with publications [22] that demonstrated that some Cannabis extracts enriched with specific cannabinoids were more potent agonists of TRPA1, TRPV1, TRPV2, and TRPM8 receptors compared to these same pure phytocannabinoids individually” may require some clarification since it is difficult to deduce what the authors mean when they talk about mechanisms without the intervention of receptors. Do they refer to enzyme inhibition? The results probably show that, in addition to CB1 and CB2, the TRP channels are very involved, since cannabichromene, in addition to being an agonist of the CB2 receptor, is a modulator of these channels.

A table with all the abbreviations is essential. Some of these are obvious, but most are not.

It is a paper I really like. It seems to me that once the table has been incorporated and some clarifications have been made, this paper deserves to be published without any other modifications.

Author Response

Reviewer 2

The sentence “Yet, CBD and CBC may act together by alternative pathways independent of receptor mediation” is not clear. A cannabinoid can be an agonist of a receptor, two at the same time not, they will compete and act depending on the affinities and concentrations according to the law of mass action. I imagine the authors mean that there are phytocannabinoids in the extract that act on different receptors. Do the authors have more information than what appears in reference 22?

Au: The indicated sentence meant to convey the possible mechanism through which CBD and CBC in the extract lead to the death of the HNSCC cell lines may be due to affecting two different signalling pathways rather than CBC acting by enhancing CBD’s activity. We added a reference to an additional paper (Muller et al., 2019) discussing the role of TRP channels as cannabinoid receptors and we clarified this sentence.

The sentence “The suggested combination of CBD-CBC in this research aligns with publications [22] that demonstrated that some Cannabis extracts enriched with specific cannabinoids were more potent agonists of TRPA1, TRPV1, TRPV2, and TRPM8 receptors compared to these same pure phytocannabinoids individually” may require some clarification since it is difficult to deduce what the authors mean when they talk about mechanisms without the intervention of receptors. Do they refer to enzyme inhibition? The results probably show that, in addition to CB1 and CB2, the TRP channels are very involved, since cannabichromene, in addition to being an agonist of the CB2 receptor, is a modulator of these channels.

Au: We were referring to polypharmacology effects, where whole cannabis extracts may be more potent than individual cannabinoids, and even greater than their additive effect (i.e., synergy), possibly due to hitting more than one target. As we did not show direct evidence for the involvement of CB1/2 and TRPV1, we refrained from making specific suggestions in the Discussion section. We did revise this sentence for clarification.

A table with all the abbreviations is essential. Some of these are obvious, but most are not.

Au: A glossary of all abbreviations was added to the paper.